# Assessing the Mechanical Properties of a New High Strength Aluminum Hybrid MMC Based on the ANN Approach for Automotive Application

**DOI:** 10.3390/ma15062015

**Published:** 2022-03-09

**Authors:** Akhileshwar Nirala, Shatrughan Soren, Navneet Kumar, Yogesh Shrivastava, Rajeev Kamal, Abdullah Ibrahem Al-Mansour, Shamshad Alam

**Affiliations:** 1Department of Fuel, Minerals and Metallurgical Engineering, Indian Institute of Technology (ISM), Dhanbad 826004, India; ssoren@iitism.ac.in; 2Galgotias College of Engineering and Technology, Greater Noida 201306, India; navneet.kumar@galgotiacollege.edu (N.K.); yogesh.shrivastava@galgotiacollege.edu (Y.S.); 3Energy Systems Laboratory, Texas A&M Engineering Experiment Station, Texas A&M University, College Station, TX 77807, USA; rajeev@tamu.edu; 4Department of Civil Engineering, College of Engineering, King Saud University, Riyadh 11421, Saudi Arabia; amansour@ksu.edu.sa (A.I.A.-M.); salam@ksu.edu.sa (S.A.)

**Keywords:** aluminum matrix composites, nanocomposite, carbon nanotube, composite materials, wear resistance, artificial neural network

## Abstract

Aluminum-based composites with characteristics such as low density and high strength to weight ratio have been identified to be one of the best-emerging alternatives. The lightweight composite is gaining popularity, particularly in the automotive industry. The composite’s qualities make it a prospective material to replace significant materials that are now used in the automobile industry. For lightweight products, various weight reduction solutions were proposed. In the present work, one such lightweight composite was fabricated by using a stir casting process, which includes reinforcement powders viz. carbon nanotube and fly ash to pure aluminum. The use of fly ash helps in reducing the overall associated cost of the material as well as provides low density. The work aims to identify the amount of fly ash (by weight %) suitable to avail good mechanical properties. In concern with the mechanical properties, density, yield strength, ultimate tensile strength, and wear resistance of the composite specimen were examined. Moreover, the artificial neural network was adopted to identify minimum volumetric wear for a given set of conditions. From the results, it was perceived that with the increase in fly ash content, the volumetric wear of the fabricated composite decreases. However, with the increase in load and speed, the volumetric wear rate increases.

## 1. Introduction

A composite consists of two or more different metal/alloys/non-metals, which have better properties than these individual materials, and have different physical and chemical properties than that individual component. Fly ash has several trade-off advantages, including being one of the most cost-effective and low-density reinforcing materials accessible as a solid waste byproduct from coal combustion in thermal power plants. Aluminum matrix composites containing fly ash particles as reinforcement are anticipated to break through the cost barrier in a variety of applications, including engineering, automotive, and others. Various investigations on the physical and mechanical characteristics of aluminum matrix composites using fly ash as reinforcements were conducted by some researchers. The lighter weight improved the engine efficiency of the automotive industry and contributed to fuel economy [1,2,3,4,5,6,7]. The growing demand for metal matrix composites is to be cost-effective and at the same time offer an optimal stage of performance experience. For a couple of decades, considerable effort has been made to develop metal matrix composite (MMC).

Several authors reviewed much progress work in casting for 50 years from 1965 and reported fabrication and the recent application of the aluminum metal matrix composite (AMMCs). In order to produce metal matrix nanocomposites, syntactic foams, self-lubrication, and auto-healing castings were presented and discussed regarding future aspects of the AMMC with effects of reinforcement such as silicon carbide, graphite, alumina, fly ash, etc. [8,9]. Some of the researchers also added magnesium as an additive to increase the wetting properties, reduce the gas layers, and obtain minimum porosity in casting products [10]. Infiltration of SiC preform in the aluminum melt along with water glass and Fe_2_O_3_ as a binder powder were used to increase wettability to fabricate a composite. This fabrication technique is new, but it is much more expensive [11,12]. Shen et al. prepared a SiC reinforced magnesium matrix composite and studied the effect of grain morphology of matrix and reinforcement in composites. They found that matrix grain size decreases by decreasing the reinforcement in the composite [13].

Ajit et al. adopted the same stir casting method to produce MMCs, and they also used reinforcement such as fly ash (type-A and type-B) and LM6 (Al-12.2 wt.% Si-alloy) as the matrix materials to strengthen MMCs. Mechanical and physical properties improved in type-B fly ash added to composite, and this is happened due to morphological differences and the existing carbon in a few amounts in it [14,15]. Some other authors also used stir casting methods to fabricate the composite along with matrix (Al6063) and reinforcement (fly ash). The mechanical properties such as hardness, tensile, and wear resistance are enhanced by the addition of fly ash in the matrix [16]. The fly ash reinforcement was used in the aluminum–copper matrix by some authors and dispersed up to 15%, and they found that wear resistance improved by the reinforced addition [7,17,18,19,20]. Hebbar et al. used an aluminum matrix with fly ash reinforcement, and they studied tensile, compression, and hardness properties and found that these properties are improved by the addition of fly ash in the matrix phase [6]. Shanmugasundram et al. also added fly ash up to 20 wt.% in the aluminum matrix and reported that the addition of fly ash decreases density but increases mechanical properties, dry sliding, and damping capacity of the composite [21]. MK Surappa used fly ash reinforcement in aluminum alloy (A356) matrix and fabricated aluminum alloy matrix composite (AAMC). They performed wear tests on these varying parameters of load (10–80 N) with 12 wt.% of fly ash in the matrix phase. Their key finding of the research obtained enhanced wear properties and corrosion resistance, and it increases with increasing the fly ash content in the aluminum matrix [5].

ANN model for predicting the densification of composite Al–SiC powders with a rigid die under uniaxial compaction was developed by Hafizpour et al. The findings of the ANN model were evaluated using densification processes, i.e., particle reorganization and plastic deformation. The correct compaction condition is necessary if the strengthening particle size is to be suited to the highest density and the volume fraction depends on the compacting environment [22,23,24]. The ANN, which covers learning from previous data, called a training or learning set, and checks system performance through test data, can provide a more precise model [25]. In several research fields, such as electronics, aerospace, robotic, medical diagnostics, etc., it can be used as a prediction and modeling [26]. In the prediction of mechanical and wear characteristics of fiber-reinforced polyamide composites, Jiang et al. used artificial neural network (ANN) techniques [27]. The fatigue life of various composite materials using ANN was anticipated by Al-Assadi et al. [28]. With numerous neural network models, Yousuf et al. forecast fatigue life for the unidirectional glass fiber and epoxies [29]. In the neural network, Zhang et al. forecast a particular wear rate and coefficient of friction of the composite [30].

In the determination of the mechanical and tribological properties of composites, the size and form of reinforcement often play a significant role. The issue of the type of strengthening, such as fly ash, is to increase the hardness and tensile strength of the fabricated composite. The objectives of the present research were to investigate the influence of fly ash and CNT reinforcement in the aluminum matrix. The study includes different characteristics of the composite, such as mechanical and tribological, whereas the stir casting fabrication method was opted to fabricate the nanocomposite. Moreover, the artificial neural network was adopted to verify the experimental volumetric wear value and identify minimum volumetric wear for the given set of conditions.

## 2. Materials and Methods

### 2.1. Materials Used in the Study

The fabrication of metal matrix composites (MMCs) was aimed in the present study, where aluminum was used as a matrix material and fly ash, and CNT was used as a reinforcement. The composite specimens were prepared using a 99.5% pure aluminum matrix with fly ash and multi-wall carbon nanotube (MWCNT) reinforcement as per Table 1. Fly ash was added to the matrix in weight of 9, 12, and 15%, but CNT reinforcement was fixed. Due to obtaining sufficient desired properties, some authors added 2% CNT reinforcement [31]. The fly ash was obtained from the NTPC Badarpur, New Delhi, India, and the powder’s characteristics and chemical composition of the as-received fly ash powder are shown in Table 2 and Table 3, respectively. MWCNT (carbon nanotube) was obtained from Ad-Nano, Chennai, India, and the details are shown in Table 4. Figure 1a,b) shows the scanning electron microscopy (SEM) micrograph of fly ash and CNT reinforcements, respectively, used in the present study. Fly ash powders and carbon nanotubes (CNT) have spherical and hair-like morphologies, respectively. The morphology of the fly ash also reveals the spherical form of the fly ash particles, which are large in scale and have smooth surfaces. The current research goal involves the use of fly ash powders to achieve lower density and CNT improved grain linkage to improve mechanical properties.

The received powders were manufactured by catalytic carbon vapor deposition methods, so there is no need to cut or chopped in a certain length. The image that appears in SEM is its common form of fly ash and CNT.

### 2.2. Fabrication Method

The squeeze stir casting method is selected among the different production methods for MMC since it tends to be the easiest way to mix the homogeneity of the particles in the matrix phase according to the composition Table 1. Squeeze casting is one of the best fabrication methods to obtain better homogeneity and overcome the porosity and shrink pores to obtain better mechanical properties of the AMMCs. This method may be understood as the combination of casting and forging techniques, which can be performed under high pressure while casting. The dispersion of the reinforcement can be evenly controlled and bond-forming by applying high pressure, and pressurized conditions also influence its wettability in molten metal. The wettability of the molten mixture increased the homogeneous mixing of fly ash and CNT. This is an easy process to fabricate the nanocomposites, which provide a strong finish on the casting surface. Figure 2 shows a bottom pouring type squeeze stir casting schematic diagram setup of the furnace in the present study. The present experimental setup contains two furnaces, where both of them are mounted on the same platform. One of them is the main heating furnace, and another is a miniature furnace. These are controlled by the PID controller for on/off the furnace. The heating furnace was used to melt the aluminum matrix and another miniature furnace for preheating the reinforcement to remove the moisture content and also enhance the wetting properties of the reinforcements. In the main heating furnace, the aluminum ingots (Approx. 1.1 kg) are melted at a temperature of 800 ± 50 °C along with both of two different additives such as potassium fluorotitanate (K_2_TiF_6_) and 1 volume % of magnesium to improve wetting capacity in the aluminum matrix with reinforcements (fly ash and CNT). The reinforcements were preheated for 60 min in the miniature furnace at a temperature of 200 ± 25 °C to increase the wetting capacity and reduce the moisture of the reinforcements.

Some of the authors fabricated nanocomposites, and they used Al_2_O_3_ nanoparticles into the semi-solid mixed of A206 (Al–Cu–Alloy) with a lower Al_2_O_3_ concentration. This semi-solid mixing method can provide a powerful tool for efficiently producing a nanocomposite with a large volume.

The nanocomposites can contribute to efficient production for industrial applications of high-performance metallic nanocomposites [32]. For the development of metal matrix nanocomposites, a high amount of nanoparticles is required to disperse in the mixture, which is essential for nanocomposite. Nevertheless, it is very difficult to mix a large number of nanoparticles in metals to obtain the key nanocomposites. Nanoparticles have low wetting properties due to the high surface area of the nanoparticles, and it leads to agglomeration and clustering in the melt. For efficient integration of nanoparticles into a metal matrix, fly ash and CNT mix (prepared in a miniature furnace) were added in semi-solid mixing for lower agglomeration. Increased shear stress during the mixing phase breaks up clusters of nanoparticles due to increased metal viscosity in a semi-solid state.

In order to understand the previous fabrication method, after melting the matrix metal, it allowed forming a semi-solid, and then preheated reinforcement was mixed slowly in the melt. The stirring melted for 10–15 min in the furnace when the vortex was formed at a variable speed between 300 and 750 rpm. The motor blades with motorized up/down movement, consisting of twisted steel blades, were used for the stirring of the semi-solid molten system. Then, by opening the bottom of the main furnace, the melt was transferred to the permanent mold through a heated delivery tube (heated pathway), where squeeze casting happened. The delivery tube was preheated to 650 °C to prevent melt into solidification nucleation in the pathway. The tube pathway is connected to a hydraulic mounting piston in the permanent mold. The melt with a pressure of 150 MPa on the piston (Piston Dia = 50 mm) was pulled in the permanent mold. The melt was kept held at that pressure by the piston and cooled to room temperature.

The composite specimens were produced using a squeeze stir casting process with the composition of AF9C, AF12C, and AF15C. The composites were produced in the bottom pouring permanent mold casting furnace in the solid cylinder form of good quality, as shown in Figure 2 (longitudinal view). Cylindrical specimens are prepared (Dia = 50 mm, L = 220 mm) and cut down for Scanning Electron Microscope (SEM), wear test, and tensile specimen, as shown in Figure 2. The entire sample is made according to the standardization of ASTM in appropriate shape and volume. The tensile specimen was prepared according to ASTM E08 [33], and similarly, the wear specimen was prepared according to ASTM G95-99.

### 2.3. Mechanical Characterization

The fabricated composite specimen was machined/cut down as per ASTM E08 specification for the tensile and ASTM E9 specification for the compressive test from the cylindrical ingot and tested on the standard ultimate tensile machine (UTM). Mechanical properties are distinguished by the density of the specimen. Density indicates the available porosity in the fabricated parts. Moreover, porosity is responsible for undesired mechanical properties.

### 2.4. Pin-on-Disc Wear Tests

Tribological properties are required to study the abrasion/wear behavior of composite specimens. The pin-on-disc machine (Figure 3) was used to observe the wear rate and frictional behavior of the composite. The specimen for the wear test was prepared according to ASTM G99-95 standards. The wear specimen (pin) is made of a chosen composite, and it is fixed with a pin holder and kept in contact with a spinning disk under an applied load. The pin may have a design to mimic a particular touch, but rounded tips are often used to simplify the geometry of the contact. The ratio of frictional force to loading force on a pin is determined by the coefficient of friction. Wear tests were performed in a laboratory at normal atmospheric pressure, humidity, and room temperatures on a pin-on-disc wear test machine. The wear test was performed on varying loads as 10, 20, and 30 N for 200, 300, and 400 rpm for a duration of 1 h. EN-32 steel (case hardened steel) disc of diameter 150 mm and thickness of 8 mm is used for the sliding track of 100 mm. Sliding wear is associated with surface interactions with pin (specimen) and disc, and deformation of the materials and their removal happened by mechanical action. The specimen’s weight loss was observed at the end of each test using an electronic weighing machine of 0.01 mg accuracy. The pin was cleaned with acetone before being weighed with a simulated electronic balance for each desired cycle. The volumetric wear rate was obtained concerning time from the test, and it was calculated using Equation (1). According to the equation, the volumetric wear rate is proportional to the change in mass. The volumetric wear is also affected by density and overall abrasion time.
(1)$$\mathrm{Volumetric}\;\mathrm{wear}\;\mathrm{rate}=\frac{mass\;loss}{density\;*\;total\;time}=\frac{\Delta m}{\sigma t}$$
where Δm is the change in mass before and after the test in kg, *σ* is the density of the nanocomposite in kg/m^3^, and *t* is the total time of wear with a disc in second.

### 2.5. ANN Modeling

Artificial neurons’ key components are weights, summing and activation function, and outputs, which are represented in Figure 4. Input data from external environments or other artificial neurons can be collected by the agent. The volumetric wear of the material was modeled by ANN in the present study. The input and the output were considered a weight fraction of fly ash, applied load and RPM of the sliding disc, and volumetric wear, respectively.

## 3. Experimental Results and Discussion

### 3.1. Microscopic Properties

The specimens Figure 2 (Dimension of SEM specimen D = 8 mm and L = 8 mm) of composites were prepared for the analysis of microstructures. Figure 5a–c shows a scanning electron micrograph of the AF9C, AF12C, and AF15C, respectively. The surfaces of the specimen were polished with a slurry of 0.1 μm diamond suspended particles. Backscattered (BS) and secondary electron (SE) were used for SEM images. SEM images show homogeneous mixing in the entire volume of the composite. Some researchers reported that uniform distribution or the addition of nanoparticles is not possible in a simple manner. However, some of them have found a means of dispersing nanoparticles into matrix materials so that lightweight composite gain good strength [32]. The volume fraction of the fly ash increased by increasing the reinforcement in the composite. Fly ash is mixed homogeneously to the entire volume, but some of them show agglomeration because the agglomeration cannot be stopped completely, and thus compositional differences from the core to the surface of the composite were found. Figure 5c depicted the AF15C composite; the addition of fly ash reduces particle size while increasing porosity. The traces of the fly ash and CNT were found, but CNT is invisible due to its higher surface area (nanosize). The interaction of fly ash with aluminum shows better in all compositions. Figure 5a–c contains 9, 12, and 15 wt.% of fly ash together with 2 wt.% of CNT, and SEM images show the same evidence.

Fly ash and CNT were observed at the grain boundary and shown in Figure 5 with an arrow. Grain boundary agglomeration inhabit for grain boundary sliding is good for enhanced mechanical properties. The segregation of fly ash and CNT was confirmed by EDS analysis in Figure 6. Spectrum 42 was captured in between the grain and spectrum 42 captured on the grain, and it was found that reinforced fly ash is agglomerated on the grain boundary, which is responsible for enhancing mechanical properties.

### 3.2. Density of Composite

Figure 7 shows the variation in measured density along with the porosity of the fabricated composite. These properties varied according to fly ash addition in the composite. Density decreases by the addition of the fly ash addition in the matrix phase. This is due to the comparatively low bulk density of the fly ash compared to aluminum, but the theoretical density of the fly ash is higher than aluminum. Theoretical density was calculated by the law of mixture. The measured density of the composite is always lower than the theoretical density. The highest density was observed in the case of lower reinforcement and minimum at higher reinforcement of the fly ash in the composite. The Archimedean method was used to evaluate the density and porosity of the composite [34,35,36].

### 3.3. Tensile and Compressive Strength

Figure 8 shows the variation in the theoretical strength, ultimate tensile strength (UTS), yield strength, and % elongation with the variation in reinforcements. The atomic bonds of a solid elastically expand when it is under strain. When a critical strain is achieved, all atomic bonds in the fracture plane shatter, and the material mechanically collapses. The theoretical strength, commonly represented as σth, is the stress at which a solid fractured. The theoretical strength (σth) can be calculated using Equation (2) [37]. The stretched atomic bonds revert to their original state after fracture, except for the formation of two surfaces. Theoretical strength increases with the addition of reinforcement because fracture stress is enhanced.
(2)$$\sigma \mathrm{th}≅\frac{E}{2\pi }\;$$
where σth and E are theoretical strength and Young’s modulus, respectively.

Tensile and yield strength increases with increasing the reinforcement by more than double the pure aluminum, but strain reduces by 6% with reinforcement. AF15C shows maximum ultimate tensile and yield strength. Reduced strain indicates the brittleness properties of the composites, and it is increasing by increased fly ash and CNT reinforcement in the aluminum matrix. These mechanical properties of nanocomposite are enhanced due to the combined effect of CNT and fly ash reinforcements. The addition of reinforcement affects elongation %, and AF15C has the least amount of elongation. The elongation of pure aluminum and maximum reinforced composite is 18.3% and 12.3%, respectively. 

The variation in ultimate compressive strength (UCS) concerning reinforcement such as fly ash and CNT in the aluminum matrix of nanocomposite is shown in Figure 9. It is increased by increasing the reinforcements in the aluminum matrix. The rate of UCS enhanced gradually high up to AF9C, but after that, it is increased comparatively low rate. For composites, the ductility of aluminum metal is an important variable that affects the elongation of the composites. It depends on the interaction between aluminum and reinforcements and the volume fraction of the reinforcements in the composite. The elongation is normally calculated for the ductility of the aluminum-based composites. The addition of fly ash in the aluminum matrix increases porosity and results in decreases in ductility, but tensile strength and yield strength increases. It is due to the fly ash hindering grain boundary sliding when crack initiates. With the presence of fly ash particles in the aluminum matrix, the grain boundary does not slip easily between the aluminum and the fly ash at the crack interface that leading to a higher tensile load. The attached fullerene modules of CNT in composite materials act as molecular anchors stopping the nanotubes from slipping and thus enhancing the mechanical properties of the composite [38]. Throughout the case of composites, rough and brittle hardened particles of fly ash are reinforced in the ductile aluminum matrix and hinder the material movement of the product, contributing to higher tensile properties.

### 3.4. Fractograph of AMMC

Figure 10 is a fractography as shown after fracture of the tensile sample AF9C, AF12C, and AF15C produced by the squeeze stir casting method. Material failure occurs due to grain boundary sliding and is initiated by microporosity followed by a pore path. The fracture of the cup and cone type was observed in the fractography. Figure 10 clearly shows the variation in ductility with the composition, and the dimple/round cleavage structure found the lowest in the case of AF15C concerning sharp cleavage. Sharp cleavage indicates the brittle properties of the composites.

Sharp cleavage increases by the addition of fly ash to the aluminum matrix. The ductility can be easily differentiated by analyzing the material of the rounded cleavage in the fractured surface.

### 3.5. Hardness

Hardness is one of the essential mechanical characteristics and is appropriate for impenetrable surfaces. Since the current research focuses on composite wearing behavior, the hardness check is important. In order to study the hardness of the composite materials, the Brinell hardness measurement was employed. Brinell hardness measuring instruments of 10 mm diameter and 250 kg load were used, and it took 30 s to load. For each sample, three measurements were taken, and the mean value of that was taken into account.

Figure 11 shows the variation in hardness with fly ash content in the nanocomposite, and it was observed that reinforced composite has higher hardness than unreinforced composite. The findings of a compression experiment indicate that introducing fly ash reinforcement increases the hardness extensively. The hardness was enhanced due to the surface morphology by the addition of fly ash.

Therefore, it can be inferred that the inclusion of fly ash and CNT enhances mechanical properties such as hardness, tensile strength, and compressive strength. Mechanical characterization such as density, tensile strength, yield strength, % elongation, and hardness tests for metal matrix composites was tested through extensively planned laboratory experiments to simulate service conditions as closely as possible. It was adequately established that the composite density was diminished and the hardness improved. Significantly, there was also an improvement in tensile strength, but there was a decrease in % elongation of the hybrid metal matrix composites. The mechanical properties such as tensile and compressive strength were increased by the addition of reinforcement of fly ash in the aluminum matrix, and it happened due to creating obstacles in dislocation movement in the composite.

### 3.6. Wear Properties

The stir casting process successfully prepared the aluminum metal matrix reinforced with 9, 12, and 15 wt.% fly ash and 2 wt.% CNT and the wear specimen has prepared as per ASTM standard, and then wear tests of the composites were studied using a pin-on-disc system. Wear involves the gradual loss of materials or damage to the outer layers of the solid. The wear is caused by plastic deformation and the removal of particles that create wear debris from the surface, as illustrated in Figure 12. The track width/thickness wear debris of the specimen varies with surface hardness and is felt as width and debris decrease with the addition of fly ash, and it is only caused by a harder surface. The surface hardness of a composite can be affected by the manufacturing process and the reinforcement used.

Figure 13 shows the volumetric wear according to applied load and RPM values for different compositions; the effect of load on wear is dominant. The volumetric wear increased as the load and sliding velocity (RPM) increased. At 20 N load and 200 RPM, the comparative wear loss was found to be high. Similarly, for minimal wear loss, a 20 N load is dominant at 300 and 400 RPM. In all RPM segments, the maximum wear loss was observed for a 30 N load. The experimental results show that RPM is the most important factor in volumetric wear loss. The materials break tangentially on the surface of the specimen due to the impact of load (vertical) and RPM (horizontal force). When RPM is increased while the load remains constant, the horizontal force is increased, causing the resultant force to be shifted towards the surface (horizontal plane) and materials erosion to become easier.

The volumetric wear decreases as the fly ash content increases, but the highest difference in wear loss is seen from AF9C to AF12C. The wear rate decreases from AF12C to AF15C. Since the slope of the graph from AF9C to AF12C is greater than the slope of the graph from AF12C to AF15C in both load and RPM values. As a result of analyzing the experimental data, it is possible to conclude that AF12C is more economical than AF15C. The variation in materials loss happened due to fly ash and CNT reinforcement in the matrix phase. Wear rate is the combined effect of the hardness and strength of the materials. Surface erosion is controlled by the hardness of the composite. Fly ash reinforcement increases hardness, and a harder surface acts as a hindrance to abrasion. Reinforcement inhibited grain sliding on grain boundaries, resulting in a high friction coefficient. Moreover, to identify the effect of fly ash content on the wear property of the composite, mathematical modeling was performed, as discussed in the ensuing section.

## 4. Artificial Neural Network (ANN) Modeling

The wear properties of the composite material are usually affected by the percentage of fly ash content in the composite. In order to identify the percentage of fly ash and wear rate at corresponding load and revolution per minute (RPM), an artificial neural network (ANN) was used [39]. With the help of ANN, a mathematical model was developed. The model is capable of predicting the variation in wear with the change in fly ash content, load, and RPM. The architecture used for generating the model is shown in Figure 14. The architecture consists of three inputs viz. percentage of fly ash (9%, 12%, 15%), load (10 N, 20 N, 30 N), and RPM (200, 300, 400). Two hidden layers have 10 and 5 neurons, respectively, and one neuron at the output layer, i.e., volumetric wear. The selection of the number of hidden layers and neurons was performed based on the Heaton theory [36]. The experimental data evaluated at different combinations of percent fly ash, load, and RPM are shown in Table 5. From the data of 27 experiments, 23 data were trained, and the remaining 4 data were tested using the feed-forward backpropagation algorithm.

For successful training of the data, the data were first normalized using the expressions given in Equations (3)–(5). The experimental combinations obtained after normalizing the data are shown in Table 6.
(3)$${\left(\mathrm{percent}\;\mathrm{of}\;\mathrm{flyash}\right)}_{\mathrm{normalized}}=\frac{\mathrm{percent}\;\mathrm{of}\;\mathrm{flyash}}{15}$$
(4)$${\mathrm{load}}_{\mathrm{normalized}}=\frac{\mathrm{load}}{30}$$
(5)$${\mathrm{RPM}}_{\mathrm{normalized}}=\frac{\mathrm{RPM}}{400}$$

### Results and Discussions

The weights obtained for different layers were obtained using MATLAB software “nntool” toolbox. Table 7 contains the weights generated from the input layer to the output. Table 8 contains the weight between the hidden layers, and Table 9 consists of the weight obtained between the output and the hidden layer. Figure 15 represents the obtained R-values for training and testing. From the figure, it is clear the data training and testing are appropriate.

Using the mathematical analogy given by Shrivastava et al. [40,41]. The mathematical model was generated. By using the mathematical model, contour plots were drawn, as shown in Figure 15. These plots show the variation in wear concerning the input parameters.

By using the weights, a mathematical model was generated, and the expression used for developing the mathematical models is as follows [42].

Initially, for developing the model, it is required to nomenclate the different layers and neurons. In the present analysis, the input layer was termed as “I” with three neurons, I_1_, I_2_, and I_3_. The second layer, which is hidden layer 1, was termed “H1”, and all the neurons in this layer were represented by H11 to H20. The third layer, which is hidden layer 2, was referred to as “H2”, and the corresponding neurons as H21 to H25. The last layer was named “O”, and the associated neuron was named O1. 

The expressions for calculating data transferred to each neuron in the first hidden layer is given by
(6)$$\{\begin{array}{l} IH11=W_{\left(input1\;to\;neuron\;H11\right)}\times F+W_{\left(input2\;to\;neuron\;H11\right)}\times L+W_{\left(input3\;to\;neuron\;H11\right)}\times R+B(I1H11)\\ .\\ .\\ .\\ .\\ .\\ .\\ IH110=W_{\left(input1\;to\;neuron\;H110\right)}\times F+W_{\left(input2\;to\;neuron\;H110\right)}\times L+W_{\left(input3\;to\;neuron\;H110\right)}\times R+B(I1H110) \end{array}\}$$

Moreover, for calculating the value of neurons in the first hidden layer, Equation (7) was used:(7)$$\left\{\begin{array}{l} H11=\frac{2}{\left[1+\exp(-2\times IH11)\right]}-1\\ .\\ .\\ .\\ .\\ .\\ .\\ .\\ H110=\frac{2}{\left[1+\exp(-2\times IH110)\right]}-1 \end{array}\right\}\;$$

Similarly, for the second hidden layer, the expression is given by Equation (8):(8)$$\{\begin{array}{l} H1H21=W_{\left(H11\;\mathrm{to}\;H21\right)}\times H11+W_{\left(H12\;\mathrm{to}\;H21\right)}\times H12+.......+B_{H1-H2}\\ .\\ .\\ .\\ H1H25=W_{\left(H11\;\mathrm{to}\;H25\right)}\times H11+W_{\left(H12\;\mathrm{to}\;H25\right)}\times H12+.........+B_{H1-H2} \end{array}\}$$

Moreover, for calculating the value of neurons in the second hidden layer, Equation (9) was used:(9)$$\left\{\begin{array}{l} H21=\frac{2}{\left[1+\exp(-2\times H1H21)\right]}-1\\ H22=\frac{2}{\left[1+\exp(-2\times H1H22)\right]}-1\\ H23=\frac{2}{\left[1+\exp(-2\times H1H23)\right]}-1\\ H24=\frac{2}{\left[1+\exp(-2\times H1H24)\right]}-1\\ H25=\frac{2}{\left[1+\exp(-2\times H1H25)\right]}-1 \end{array}\right\}$$

The expressions for calculating “*O*” was given by Equation (10):(10)$$\begin{array}{l} O1=W_{H21\;\mathrm{to}\;\mathrm{output}}\times H21+W_{H22\;\mathrm{to}\;\mathrm{output}}\times H22+W_{H23\;\mathrm{to}\;\mathrm{output}}\times H23r\\ +W_{H24\;\mathrm{to}\;\mathrm{output}}\times H24+W_{H25\;\mathrm{to}\;\mathrm{output}}\times H25+B_{H2-O} \end{array}$$

The mathematical model for normalized volumetric wear was obtained as Equation (11):(11)$$Volumetric{\;\mathrm{wear}}_{\left(\mathrm{normalized}\right)}=\frac{2}{\left[1+\exp(-2\times O1)\right]}-1$$

The developed model shown in Equation (10) was used to predict the variation in wear with the change in input parameters. Figure 16 shows three plots for different combinations of input parameters and output. For plotting these graphs, two parameters at a time were considered by fixing the third parameter. From the plot, it is clear that the region in red and yellow color has a minimum value of wear, while the blue and violet regions have maximum wear. Hence, it is suggestable to select a combination of input parameters that may result in the value of wear within the red or yellow region.

Following the above-mentioned criterion, the authors developed a range of input parameters for every plot. The range is listed in Table 7. Table 7 also contains a combined range of input parameters effective for all the three plots simultaneous, prominent in achieving minimum wear rate.

From Table 10, it is clear that for the given range of input parameters if the load and RPM are adjusted between 15.5 and 23 and 200 and 375, respectively. Then the variation in % the weight of fly ash can be of any amount. Or, in other words, for the maximum content of fly ash (15%) in the composite, the wear is minimum when the load is between 15.5 and 23 and RPM is between 200 and 375. Moreover, in order to verify the obtained range, more experiments were performed, and the wear was measured. The combination of load and RPM selected for experimentation was listed in Table 11. Moreover, the calculated volumetric wear is listed in the table. From the verification results, it was revealed that the obtained common range is significant.

## 5. Conclusions

The aluminum hybrid nanocomposite was fabricated via mixing of fly ash and CNT reinforcement by the squeeze stir casting method. The aim was to decrease the density of material with minimum to no compromise in mechanical properties and to minimize the associated cost. From the analysis, it was observed that fly ash is used as a reinforcement and has enhanced mechanical or wear properties of the MMNCs. Composites from aluminum metal matrix were fabricated using squeeze stir casting methods with different amounts of fly ash and CNT. The aluminum stir casting route to manufacture composites can be successfully added up to 15% by weight. With the added fly ash and CNT, the hardness of aluminum nanocomposite increased from 73 BHN to 89 BHN. The ultimate strength of the traction improved with an increase in the amount of fly ash. Compressive strength improves from 485 MPa to 512 MPa with reinforcement. Tensile strength is enhanced by 36% of the unreinforced composite. Experimental data of volumetric wear verified with predicted value and optimum design parameter is considered suitable for better performance. Moreover, ANN was used to monitor the variation in volumetric wear in the fabricated composite specimen. From the result, it was found that, from the maximum content of fly ash (15%) in the composite, the wear is minimum when the load is between 15.5 and 23 N, and RPM is between 200 and 375.

## Figures and Tables

**Figure 1 materials-15-02015-f001:**
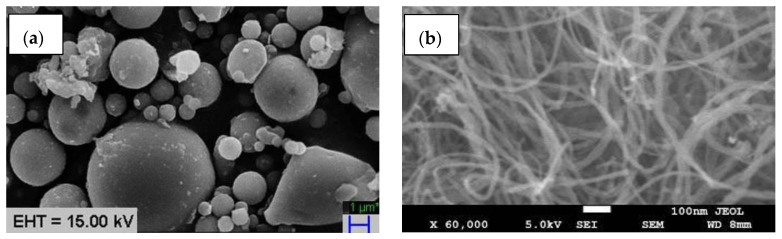
SEM images of the reinforcements: (**a**) fly ash and (**b**) CNT.

**Figure 2 materials-15-02015-f002:**
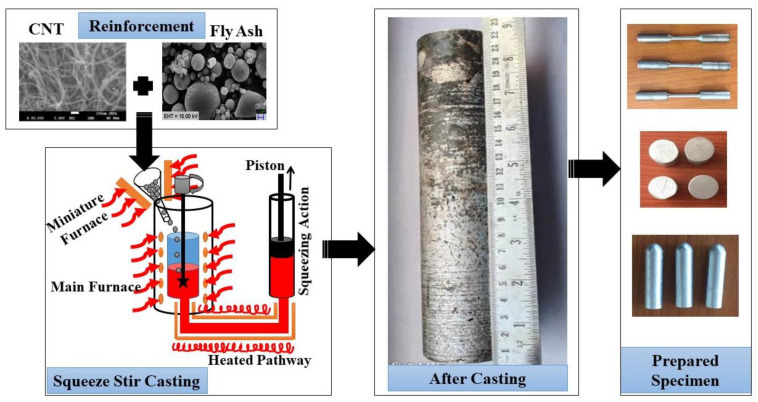
Process flow diagram for the composite fabrication.

**Figure 3 materials-15-02015-f003:**
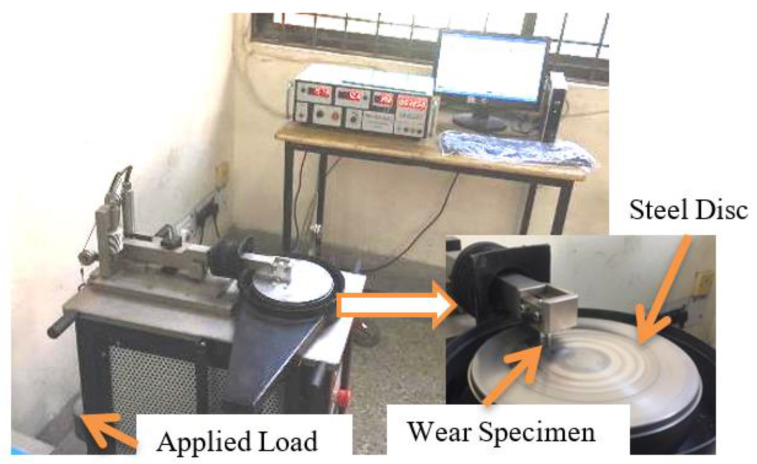
Photograph of pin on disc machine.

**Figure 4 materials-15-02015-f004:**
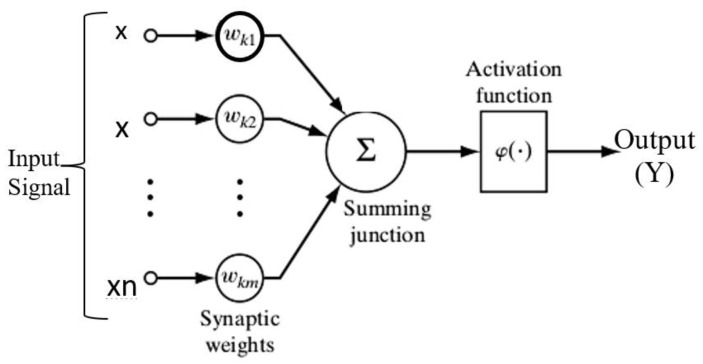
Simple ANN Architecture used in the study.

**Figure 5 materials-15-02015-f005:**
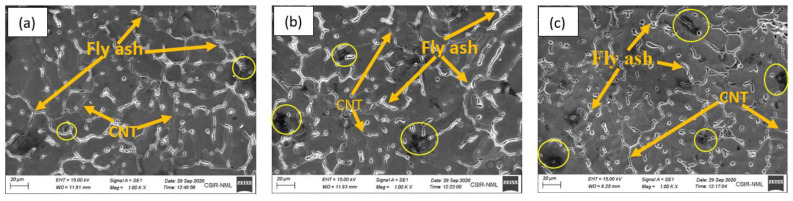
SEM micrographs of different specimens: (**a**) AF9C, (**b**) AF12C, and (**c**) AF15C and Porosity (Circle).

**Figure 6 materials-15-02015-f006:**
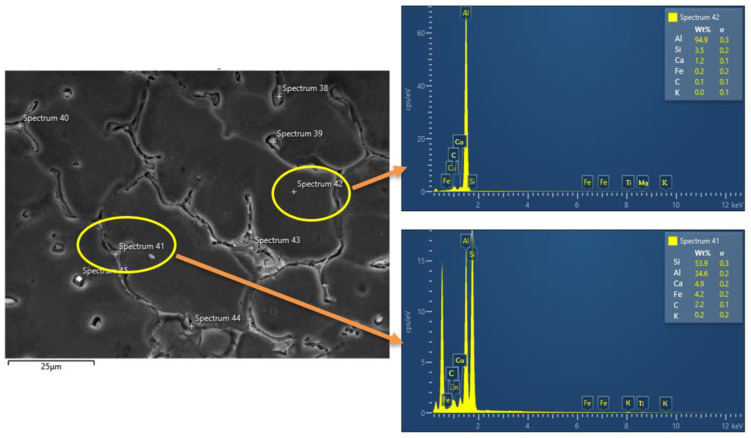
EDS analysis at grain boundary (Spectrum 41) and grain interior (Spectrum 42) of the composite.

**Figure 7 materials-15-02015-f007:**
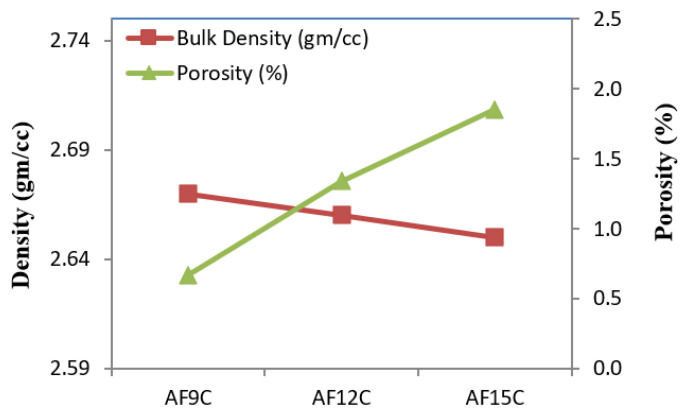
The measured density and porosity of nanocomposite.

**Figure 8 materials-15-02015-f008:**
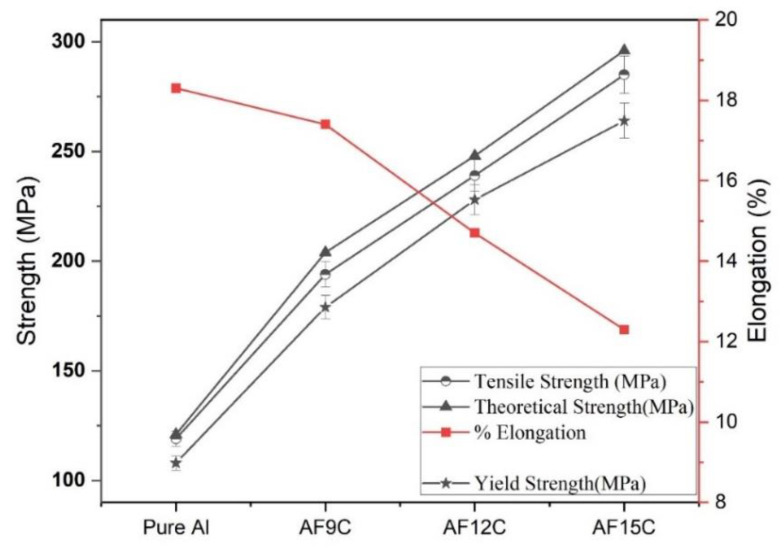
Variation in theoretical strength, tensile strength, yield strength, and strain of the composite.

**Figure 9 materials-15-02015-f009:**
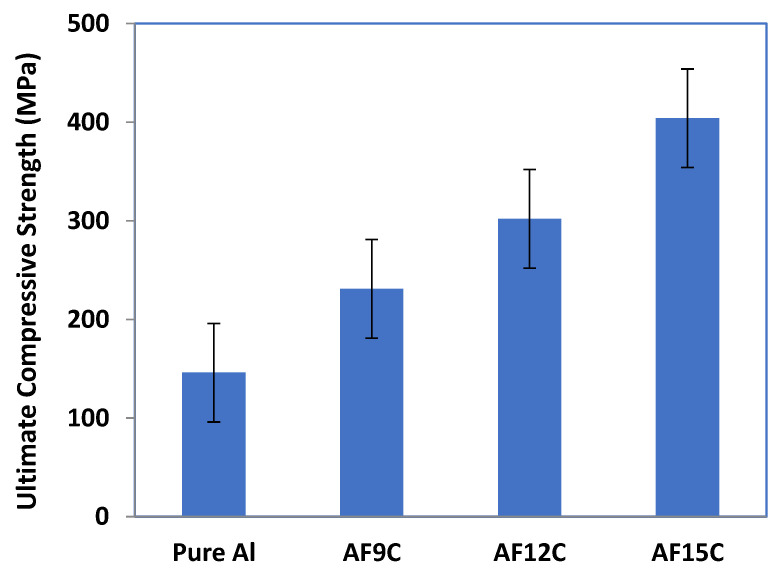
Variation in compressive strength of the aluminum nanocomposite.

**Figure 10 materials-15-02015-f010:**
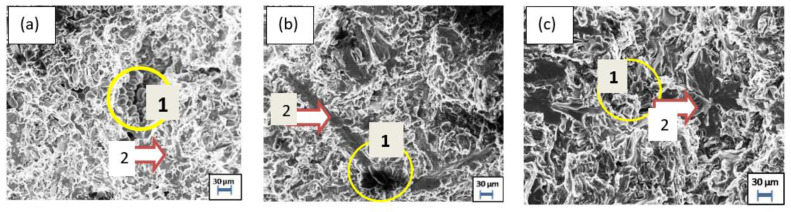
Fractographs of different specimens: (**a**) AF9C, (**b**) AF12C, and (**c**) AF15C (1(circle)—Ductile and 2(arrow)—Brittle fracture).

**Figure 11 materials-15-02015-f011:**
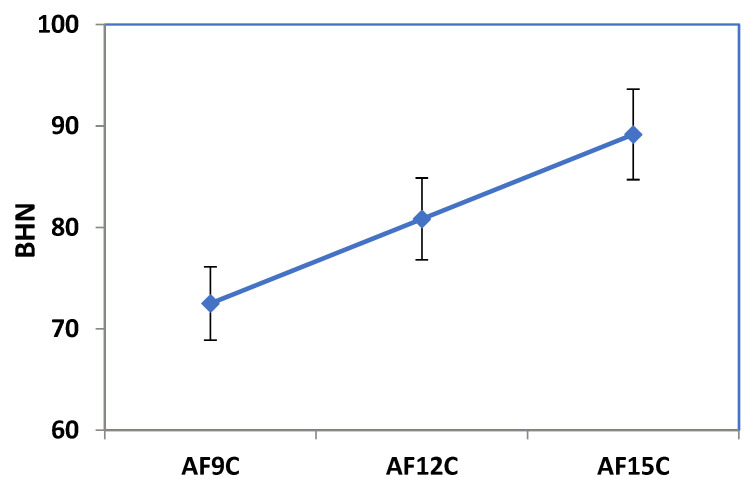
Hardness value of the different composites.

**Figure 12 materials-15-02015-f012:**
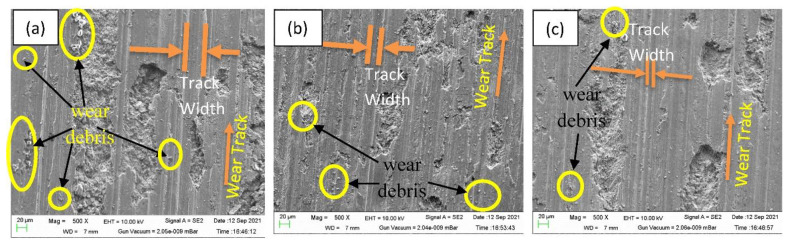
Wear track and width morphologies of different specimen: (**a**) AF9C, (b) AF12C, and (**c**) AF15C.

**Figure 13 materials-15-02015-f013:**
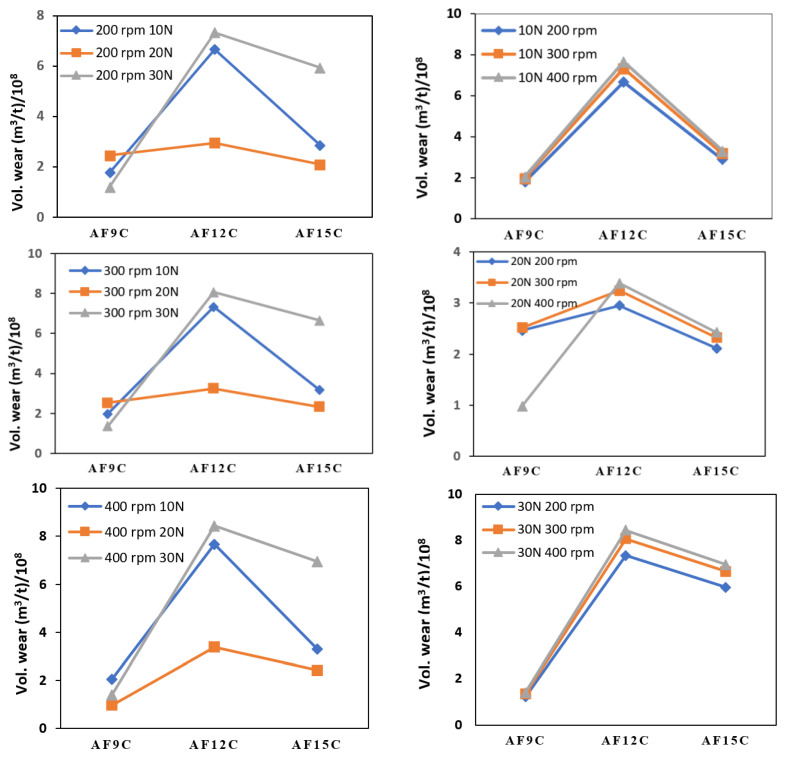
Effect of Disc RPM and Load Variation on volumetric wear of the composite specimen.

**Figure 14 materials-15-02015-f014:**
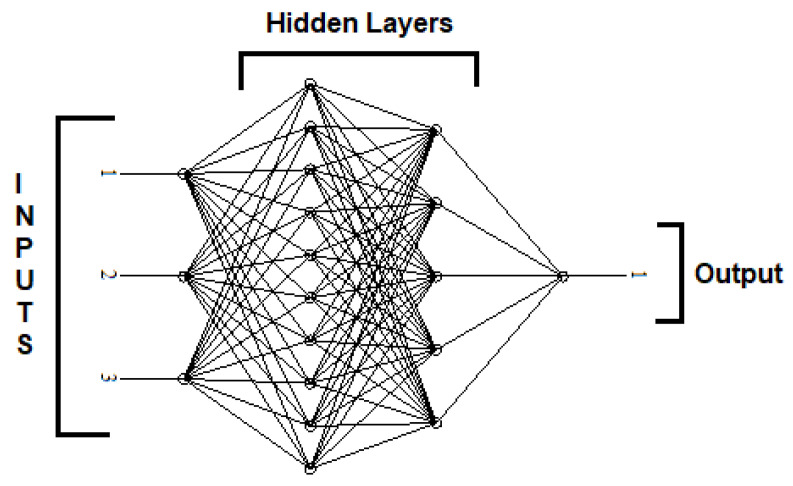
ANN architecture (3(inputs)-10(hidden layer)-5(hidden layers)-1(output)).

**Figure 15 materials-15-02015-f015:**
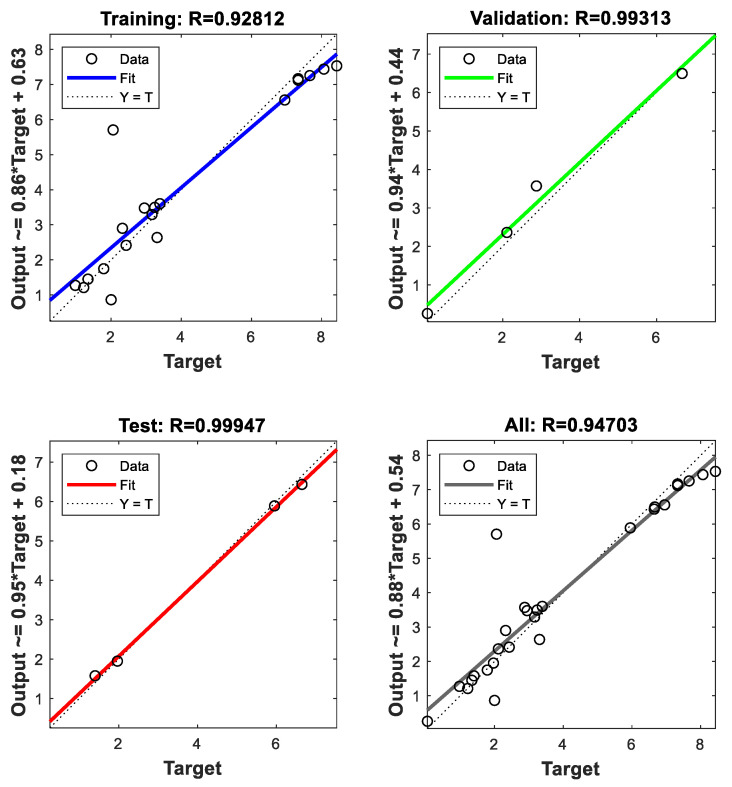
R-values for training and testing data sets.

**Figure 16 materials-15-02015-f016:**
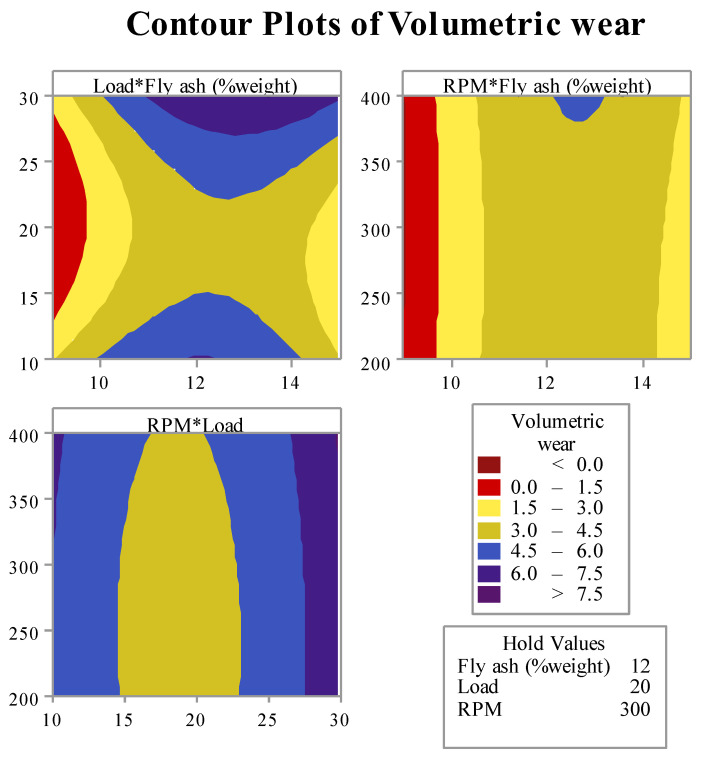
Variation in wear w.r.t. input parameters.

**Table 1 materials-15-02015-t001:** Composite Nomenclature used in the experiment.

S. No.	Composite Composition (in Weight %)	Nomenclature
1	Al (89) + Fly ash (9) + CNT (2)	AF9C
2	Al (86) + Fly ash (12) + CNT (2)	AF12C
3	Al (83) + Fly ash (15) + CNT (2)	AF15C

**Table 2 materials-15-02015-t002:** Characteristics of fly ash powder used in the study.

Property	Powder
Powder shape	Spherical
Cumulative size, μm	
D_10_	8.81
D_50_	41.01
D_90_	47.43
Specific surface area, m^2^/g	2.62
Theoretical density, g/cm^3^	2.9

**Table 3 materials-15-02015-t003:** Chemical properties of the as-received fly ash powder.

Component	SiO_2_	Al_2_O_3_	Fe_2_O_3_	TiO_2_	K_2_O	MgO	CaO	Na_2_O	LOI
% Dry weight	63.4	23.7	1.78	2.44	2.9	2.35	0.9	0.65	1.88

**Table 4 materials-15-02015-t004:** Characteristics of MWCNT used in the study.

Property	Description
Type	Multiwall Carbon Nanotubes (MWCNT)
Color	Black Powder
Purity	>99%
Average Diameter	10–15 nm
Average Length	~5 μm
Amorphous carbon	<1%
Surface Area	~400 m^2^/g

**Table 5 materials-15-02015-t005:** Actual experimental data.

Percentage of Flyash (F)	Load (L)	RPM (R)	Vol. Wear (m3/t)/108
9	10	200	1.7887
12	10	200	6.6656
15	10	200	2.8782
9	20	200	2.4600
12	20	200	2.9474
15	20	200	2.1111
9	30	200	1.2184
12	30	200	7.3324
15	30	200	5.9532
9	10	300	1.9675
12	10	300	7.3321
15	10	300	3.1660
9	20	300	2.5200
12	20	300	3.2421
15	20	300	2.3222
9	30	300	1.3400
12	30	300	8.0654
15	30	300	6.6490
9	10	400	2.0570
12	10	400	7.6650
15	10	400	3.3099
9	20	400	0.9780
12	20	400	3.3895
15	20	400	2.4277
9	30	400	1.4012
12	30	400	8.4313
15	30	400	6.9516

**Table 6 materials-15-02015-t006:** Normalized values of Experimental data.

(Percentage of Flyash)_normalized_	(Load)_normalized_	(RPM)_normalized_	Vol. Wear/10^8^
0.6	0.33	0.5	1.7887
0.8	0.33	0.5	6.6656
1	0.33	0.5	2.8782
0.6	0.66	0.5	2.4600
0.8	0.66	0.5	2.9474
1	0.66	0.5	2.1111
0.6	1	0.5	1.2184
0.8	1	0.5	7.3324
1	1	0.5	5.9532
0.6	0.33	0.75	1.9675
0.8	0.33	0.75	7.3321
1	0.33	0.75	3.1660
0.6	0.66	0.75	2.5200
0.8	0.66	0.75	3.2421
1	0.66	0.75	2.3222
0.6	1	0.75	1.3400
0.8	1	0.75	8.0654
1	1	0.75	6.6490
0.6	0.33	1	2.0570
0.8	0.33	1	7.6650
1	0.33	1	3.3099
0.6	0.66	1	0.9780
0.8	0.66	1	3.3895
1	0.66	1	2.4277
0.6	1	1	1.4012
0.8	1	1	8.4313
1	1	1	6.9516

**Table 7 materials-15-02015-t007:** Weights and bias between the input layer and the hidden layer.

Neuron (Layer 1)	Input 1 to Neurons	Input 2 to Neurons	Input 3 to Neurons	Bias (I-H1)
H11	−1.2281	−2.9455	0.29758	2.7545
H12	2.1689	0.23498	−1.8124	−2.5783
H13	2.812	0.3342	0.72451	−1.9621
H14	−0.13732	−2.639	1.3884	0.6613
H15	0.99369	−1.1095	2.5917	−0.83144
H16	−2.2247	−2.279	−0.2731	−0.80617
H17	−2.6144	0.85577	−0.79926	−1.2932
H18	0.68642	−0.88654	−2.4912	2.1604
H19	1.8828	−0.34521	2.3484	2.4044
H20	2.2416	−1.6273	−1.4484	2.9573

**Table 8 materials-15-02015-t008:** Weight and bias between hidden layers.

Neurons (Layer 2)	Neuron H11 to Neurons	Neuron H12 to Neurons	Neuron H13 to Neurons	Neuron H14 to Neurons	Neuron H15 to Neurons	Neuron H16 to Neurons	Neuron H17 to Neurons	Neuron H18 to Neurons	Neuron H19 to Neurons	Neuron H20 to Neurons	Bias (H1–H2)
H21	−0.14273	−0.22173	1.0254	0.21519	−0.30383	0.5106	−1.2519	−0.30684	−0.63051	0.59391	−2.0712
H22	−0.42984	0.63336	−0.30736	0.93948	0.6877	1.6463	−0.99475	0.26943	−0.91802	−0.16437	−1.0556
H23	−0.009095	−0.3646	0.15936	−0.51891	0.19994	−1.26	−1.5963	0.9522	0.72564	0.5682	0.010763
H24	−1.1021	−0.40649	−0.52545	−1.0463	0.31022	0.015898	−0.39148	0.08277	−0.2331	−0.46805	−0.83138
H25	0.39389	0.62185	0.40617	−0.79483	−1.0871	0.32587	0.14223	0.41268	−0.92158	−0.89277	−1.3253

**Table 9 materials-15-02015-t009:** Weight and bias between hidden layer and output.

Output	H21 to Output	H22 to Output	H23 to Output	H24 to Output	H25 to Output	Bias (H2-O)
O1	−0.57122	0.61784	0.50638	0.80524	−0.60319	−0.44983

**Table 10 materials-15-02015-t010:** Suitable range pertaining to minimum wear.

	Hold Value	Hold Value—Load	Hold Value—RPM	Combined Range
Flyash (%weight)	-	9–15	9–15	9–15
Load	15.5–23	-	15–23	15.5–23
RPM	200–400	200–375	-	200–375

**Table 11 materials-15-02015-t011:** Experiments for verification.

S. No.	Fly Ash (%Weight)	Load	RPM	Volumetric Wear/10^8^
1.	9%	16	200	0.9149
2.	12%	20	250	0.9746
3.	15%	20	350	1.4971

## Data Availability

Data sharing is not applicable to this article.

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
