# Peer review of "Assessing the Mechanical Properties of a New High Strength Aluminum Hybrid MMC Based on the ANN Approach for Automotive Application"

_materials, 2022, doi:10.3390/ma15062015_

Round 1
Reviewer 1 Report
In this research, aluminum hybrid composites were fabricated and analyzed to reduce the material density and minimize the associated costs with minimal or no loss of mechanical properties. In their experiments, three aluminum composite samples were fabricated using a stir casting process with different amounts of fly ash and a fixed amount of CNTs.
In the abstract and summary of the manuscript, the authors emphasize the improvement of ultimate strength in traction, tensile and compressive strength, and hardness of composites without much mention of the loss of elongation. However, for the practical industrial application of lightweight composites, it is worthwhile to discuss in more detail the complex trade-off benefits of fly-ash added composite (between the strength and toughness).
In the presentation of wear properties in Fig. 13, the authors adopted the total loss of material instead of the volumetric wear rate used in the ANN model. Since the numerical trend in Table 5 looks quite different from the graph in Figure 13, using materials loss in Fig. 13 seems to be a little bit tricky if the overall abrasion time is different from sample to sample. The author needs to explain the reason for using the mass loss in Section 3.6 and the wear rate in Section 4.1 beyond a model expression.
The author utilized an ANN model to analyze the volumetric wear rates of fabricated composite samples. However, only 27 (=3×3×3) experimental data were used for model training and test in the ANN model, which appears to be of relatively small size. With such a small data set size, we think conventional multivariable regression would be much simpler, and sufficient to extract the underlying relationship between input conditions and output properties. In this regard, the author should further describe the reasons for using the ANN model, the main difference from regression analysis, and the physical significance of wear analysis.
Here are some minor suggestions.
- It would be better if the abbreviated terms such as MMC (metal matrix composites), AMMC (aluminum metal matrix composites), MMNC, were used after spelling out the full term at its first mention.
- Some paragraphs need proofreading: "equations 3-5"->"equations 3-5" in line # 428, "Table 7" -> "Table 10" in line # 493~497, "6. Conclusion" -> "5. Conclusion" in line # 509, …
Author Response
We are very much grateful to review the manuscript and the comment file has been attached.

Reviewer 2 Report
The uniform distribution of ash in aluminum is not clear. It's hard to compare from the pictures. Numerical processing of images is required (for example, by the secant method).
One of the main values ​​for the mechanical properties of aluminum is the amount of oxygen in the aluminum itself, the ash introduces a noticeable change in the structure of the grain boundaries of the matrix. I would like to see images with a high magnification of these boundaries and the oxygen concentration in the matrix.
There is no comparison of the theoretical estimate of the strength of aluminum with the one obtained by the authors, it is necessary to add it.
Photographs of mixed aluminum powders with ash and nanotubes must be provided.
The choice of 2% nanotubes by the authors is not clear.
When the authors talk about the influence of nanotubes (line 333), they do not provide references for confirmation.
Author Response
Thank you for review the manuscript and comment file has been attached.

Round 2
Reviewer 1 Report
I think most of my critics and comments have been addressed and reflected in the revised manuscript.
Since the manuscript has been well improved, I think that the paper is worth to be published.
Reviewer 2 Report
I thank the authors for the informative answer! The wish to regard the oxygen in the ash and its effect on the matrix remains.